Blockchain and explainable-AI integrated system for Polycystic Ovary Syndrome (PCOS) detection

http://orcid.org/0000-0002-3327-2710 Jaganathan Gowthami mail2gowthamij@gmail.com
Natesan Shanthi
Department of Computer Science and Engineering, Kongu Engineering College , Erode, Tamilnadu , India
Rajaraman Sivaramakrishnan
Electronic publication date: 2025 Feb 28
Publication date: 2025
Volume: 11
Electronic Location ID: e2702
Received 2024 Oct 24; Accepted 2025 Jan 23
Copyright: © 2025 Jaganathan and Natesan
Copyright year: 2025
Copyright holder: Jaganathan and Natesan
License: This is an open access article distributed under the terms of the Creative Commons Attribution License, which permits unrestricted use, distribution, reproduction and adaptation in any medium and for any purpose provided that it is properly attributed. For attribution, the original author(s), title, publication source (PeerJ Computer Science) and either DOI or URL of the article must be cited.
License URL: https://creativecommons.org/licenses/by/4.0/

Keywords: PCOS prediction, Machine learning, Hyperledger fabric, Health information exchange, Predictive analytics, Explainable AI

Funding: The authors received no funding for this work.

==============================
In the modern era of digitalization, integration with blockchain and machine learning (ML) technologies is most important for improving applications in healthcare management and secure prediction analysis of health data. This research aims to develop a novel methodology for securely storing patient medical data and analyzing it for PCOS prediction. The main goals are to leverage Hyperledger Fabric for immutable, private data and to integrate Explainable Artificial Intelligence (XAI) techniques to enhance transparency in decision-making. The innovation of this study is the unique integration of blockchain technology with ML and XAI, solving critical issues of data security and model interpretability in healthcare. With the Caliper tool, the Hyperledger Fabric blockchain’s performance is evaluated and enhanced. The suggested Explainable AI-based blockchain system for Polycystic Ovary Syndrome detection (EAIBS-PCOS) system demonstrates outstanding performance and records 98% accuracy, 100% precision, 98.04% recall, and a resultant F1-score of 99.01%. Such quantitative measures ensure the success of the proposed methodology in delivering dependable and intelligible predictions for PCOS diagnosis, therefore making a great addition to the literature while serving as a solid solution for healthcare applications in the near future.

Introduction

Individual well-being is paramount to sustain and advance the collective welfare of the family, community and nation. However, the health industry is facing several obstacles in health data storage and timely retrieval of information for predictive healthcare analytics. The hormonal imbalance condition known as Polycystic Ovary Syndrome (PCOS) is caused by an excess of hormones produced by the ovaries, which are the organs responsible for producing and releasing eggs. It is a widely prevalent reproductive hormone-related disorder. Its common disruptive symptoms are cycles of menstruation that are irregular, abnormal hair growth, pimples and reduced fertility. Women with PCOS generate more masculine hormones than normal (Alanazi, 2022; Permana, Leslie & Perdana, 2023; Rubinger et al., 2023). Most women remain ignorant of PCOS-related symptoms, effects and its associated challenges. As per the report by World Health Organization (WHO), around 8–13% of women experience this distressing condition in their reproductive years, with as many as 70% of cases going undiagnosed (Goodman & Flaxman, 2016). To properly diagnose PCOS and offer personalized care, complete clinical information and detailed medical history of the patient must be stored and available for retrieval on time. The integration of machine learning (ML) and blockchain for automated PCOS prediction is poised to greatly enhance the overall reproductive health and well-being of women.

ML has been established as a potent asset across different application domains with the development of intelligent predictive algorithms within the realm of artificial intelligence. There are three distinct categories of ML models: supervised learning, unsupervised learning, and reinforcement learning. ML has transformed nearly every industry, especially in the healthcare sector with its advancements in technologically driven solutions such as disease detection, risk prediction, medical imaging diagnosis, drug discovery, fraud detection and individualized treatment. For assisting medical professionals in arriving at a precise diagnosis of diseases based on manifesting symptoms, devising effective treatments and offering enhanced healthcare, a variety of ML technologies must be incorporated into medical practices as a decision support system (Adadi & Berrada, 2020; Abouhawwash et al., 2023). The appropriate data-efficient ML techniques must be used if these versatile real-time applications are to be effectively implemented in the healthcare sector. K-nearest neighbour (KNN), support vector machine (SVM) and artificial neural network (ANN) are few machine learning algorithms that have been frequently used in disease detection research (Alamoudi et al., 2023).

In healthcare diagnosis, even a minor error could have unintended consequences. The main drawback of ML models is the lack of transparency, i.e., conventional AI is a “black box” that can only answer questions with a straightforward “yes” or “no” without presenting a justification. The necessity of providing end users with machine learning explanations and AI transparency has gained significance with several studies. For instance, black box techniques became more challenging to utilize in applications especially after the restriction of automated individual decision-making by the law that came into effect throughout the European Union (EU) in 2018 (Goodman & Flaxman, 2016). An explanation about algorithmic decisions made based on personal data must be provided to the user and user’s rights must be safeguarded at all times (Hoffman et al., 2018).

The explainable AI market is predicted to reach a valuation of USD 24.58 billion by 2030, from 5.10 billion USD in 2022 (NMSC, 2025). In addition to many popular ML models, Explainable Artificial Intelligence (XAI) provides explanations for its forecasts, enabling people to have a better comprehension of the components that make up the forecast and bringing transparency to medical forecasting (Adadi & Berrada, 2020; Gupta & Seeja, 2024). The applications of XAI in medicine can be categorized as below, Local Interpretable Model-Agnostic Explanations (LIME), Shapley Additive Explanations (SHAP), Partial Dependence Plots (PDPs), Individual Conditional Expectation (ICE), Accumulated Local Effects (ALE) and Permutation Feature Importance (PFI) (Mohanty & Mishra, 2022; Saraswat et al., 2022).

In the present digital age, blockchain technology’s enhanced immutability, security, and decentralization have made it more and more significant in many service sectors. In the health sector, robust security is essential for promoting trust, thwarting malicious attacks, enforcing legal requirements, guaranteeing stakeholders’ well-being, preserving patient privacy and safeguarding sensitive data (Hasselgren et al., 2020; Zaabar et al., 2021; Saeed et al., 2022; Wahyuni et al., 2024). The three main issues in health database management are interoperability, data sharing, and privacy. The blockchain technology effectively addresses these issues and makes it easier for healthcare professionals and organizations to deliver high-quality medical services and individualized patient care (Haleem et al., 2021). Hyperledger Fabric, which is a private blockchain network, has been very efficient in various healthcare applications such as Electronic Health Record (EHR) management, patient consent management, secure interoperability of health information, data integrity, personal monitoring of stored data and supply chain management (Kasyapa & Vanmathi, 2024). Hence, it is deemed suitable for the suggested PCOS predictive analytics framework.

The increasing prevalence of PCOS among women globally highlights an urgent need for effective diagnostic and management solutions. With estimates suggesting that PCOS affects approximately 8–13% of women of reproductive age, early detection and tailored treatment are crucial for improving health outcome (WHO, 2019). However, traditional healthcare systems often struggle with data silos, lack of interoperability, and privacy concerns, hindering timely diagnosis and care.

In the digital era, the integration of advanced technologies such as blockchain and ML presents a transformative opportunity to address these challenges. Blockchain offers a secure and immutable way to manage sensitive health data, ensuring confidentiality and patient control over their information. Simultaneously, machine learning algorithms can analyze vast amounts of health data to identify patterns and make predictions, enabling proactive healthcare interventions.

This study is motivated by the need to create a robust, secure platform that not only protects patient data but also leverages predictive analytics to enhance the diagnosis and management of PCOS. By combining blockchain’s security with the analytical power of ML and the transparency of Explainable AI, we aim to establish a system that fosters trust among patients and healthcare providers, ultimately leading to improved patient outcomes and a more efficient healthcare delivery model.

Moreover, as developing nations accelerate their digital transformation, it is essential to equip healthcare systems with innovative solutions that are both secure and scalable. This research contributes to the growing body of knowledge on digital health technologies, providing a blueprint for future advancements in patient data management and disease prediction. Even though much previous research exists for PCOS detection utilizing many ML and deep learning methodologies, the decisions made by AI models are often opaque and unreliable. The proposed Explainable AI-based blockchain system for Polycystic Ovary Syndrome detection (EAIBS-PCOS) system integrates blockchain and XAI which ensures security, aids scalability, facilitates transparency and enhances interpretability. The major contributions of this research work are, To design a novel framework for PCOS health data storage and information access, by utilizing private Hyperledger fabric blockchain that guarantees the immutability of records and restricts access to only approved entities for network participation.

To analyse several ML models and incorporate the best and most suitable one with XAI models to improve decision-making and interpretable solutions by offering a transparent and accurate Artificial Intelligence (AI) driven PCOS diagnosis system.

To implement the designed model with a dataset and to validate the system’s efficiency with respect to throughput, latency, execution time and resource consumption to identify the optimized network. Further, the optimized network is integrated with a ML model and assessed using performance metrics like model accuracy, precision, F1-score and recall.

This research article is comprehensively structured into detailed sections. The second section, which follows the Introduction, presents the insights gathered from an extensive literature study. The article’s third section provides a thorough analysis of the suggested framework with the details of chosen ML and XAI algorithms. The experimental setup and implementation details with performance metrics are tabulated in the fourth section. The fifth part examines the results and discussions of the relevance of the experimental implementation. The sixth section reviews the main points and provides the concluding summary along with recommendations for further research.

Related works

A significant amount of research is currently being conducted in PCOS detection. This section gives a summary of the systematic literature review undertaken to obtain insights for the current study.

PCOS identification based on machine learning and deep learning

A recent study by Subha et al. (2024) concentrated on PCOS detection with the use of ML algorithms. For feature selection, Swarm Intelligence (SI) was used, whereas for disease prediction, random forest (RF) and XGBoost classifiers were used. Both Flashing Firefly (FF) and Particle Swarm Optimization (PSO) feature selection algorithms yielded 92% accuracy. In the research work by Danaei Mehr & Polat (2022), different feature selection methodologies were used on the same dataset. The model’s efficacy was analyzed with various ensemble classifiers for both feature reduced dataset and feature not-reduced dataset. This study concluded that Ensemble random forest classifier performed well along with embedded feature selection in comparison with other classifiers. A wide collection of ML algorithms had been applied on the tabular non-invasive parameters and their performances have been analyzed in Tiwari et al. (2022). Random forest was chosen as the best model with 93.25% accuracy.

Ravishankar et al. (2023) adopted a fuzzy-based convolutional neural network (CNN) method for determining the presence of ovarian cysts. This technique yielded good results with 98.37% accuracy rate. An efficient PCOS detection method was proposed by a research study found that Extra trees classifier, in combination with Genetic algorithm for feature selection, gave better results for PCOD disease detection in comparison with other classifiers (Munjal & Khandia, 2020). Nandipati, Chew & Wah (2020) performed out an analysis with RapidMiner and Python Scikit package tools using several classifications and feature selection techniques for the same dataset, they found that RapidMiner tool performed better. An app for PCOS diagnosis was developed by an earlier research study, analyzing several boosting and bagging algorithms (Kavitha et al., 2023). It was observed that the proposed web app facilitated prediction utilizing Grid search cross-validation (CV) optimization with an accuracy of 94%.

Suganya et al. (2023) used the Pelvic Computed Tomography (PCT) dataset to detect ovarian cysts. The features were extracted using GoogleNet and the output was supplied to XGBoost algorithm for further classification. When compared to deep learning networks, this approach proved to be more accurate and efficient. There were a few additional studies that focused on using deep learning’s transfer learning models like VGG16, MobileNet, Inception v3, U-Net and Fusion techniques to analyze ultrasonic imaging data for PCOS (Vikas, Radhika & Vineesha, 2021; Lv et al., 2022; Abouhawwash et al., 2023; Alamoudi et al., 2023).

Blockchain and machine learning integration

An earlier study (Hu & Kar, 2024) focused on the development of a secure health monitoring system using blockchain technology and distributed ML. Hyperledger Fabric was utilized for user management and grouping, and ModelChain was used for model training. Sharma & Rohilla (2023) developed a framework with blockchain and ML for drug discovery chain management. The system’s efficiency was improved with an increase in endorsers and block sizes, and comparative study further validated the scalability of the architecture. The system’s performance was successfully assessed using the throughput and model correctness.

A system for detecting the severity of cardiovascular disorders by combining the naïve Bayes algorithm and Hyperledger Fabric was proposed by Sasikumar & Karthikeyan (2023). Using this system, physicians could easily determine the patient’s illness severity level and quickly identify high-risk patients. The designed system outperformed the existing system based on the comparison of central processing unit (CPU) and memory utilization. Heidari et al. (2023) addressed the need for timely lung cancer detection by using blockchain-based federated learning (FL) to identify the malignant nodules. The obtained outcomes revealed that the CapsNet classification was effective in identifying lung cancer patients, with 99.69% accuracy and minimum classification errors. Pandey et al. (2022) proposed a system for waste material identification and recommendations for reducing the quantum of disposed waste with its reuse. The system utilized deep learning’s ResNet50 model for exact object identification. Upon successful object identification, the system leveraged a Web Scraper to get applicable recycling ideas from the internet. This approach encouraged environmental sustainability and facilitated environmental pollution reduction.

A reliable system for veterinary data storage and analytics was proposed by Iqbal et al. (2021) it could make several predictive analyses to improve veterinary services, using regression and deep learning models. Implemented with Hyperledger Fabric in two variations, a single Kubernetes cluster and in a cross-cluster environment, the proposed system employed advanced algorithms to accurately forecast appointment scheduling frequencies, leading to optimal resource allocation and increased operational efficiency in veterinary healthcare settings. A natural language processing (NLP) based model integrated with blockchain to convert the doctor’s prescription into text was presented by Bharimalla et al. (2021). This system used Hyperledger Fabric to build the EHR system, integrated with national identity documents like Aadhaar or PAN (Permanent Account Number) to easily identify the patient’s medical records.

The reviewed studies on PCOS detection and Blockchain-ML integration present a variety of approaches, each with its strengths and limitations. Subha et al. (2024) achieved 92.64% accuracy using Swarm Intelligence for feature selection and random forest (RF) and XGBoost classifiers but lacked blockchain integration for secure data management. Danaei Mehr & Polat (2022) have proposed Ensemble random forest classifiers that integrate feature selection from within; however, this has improved performance but doesn’t cover scalability and blockchain security. Ravishankar et al. (2023) have obtained a high accuracy of 98.37% with the fuzzy-based CNN for the detection of ovarian cysts. They propose a strong deep learning-based approach, yet they don’t involve blockchain. Suganya et al. (2023) applied GoogleNet for feature extraction and XGBoost for classification, which outperformed other deep learning models but was not equipped with data security measures. In blockchain-ML integration, Sharma & Rohilla (2023) showed how Hyperledger Fabric and ModelChain improved scalability and throughput in secure health monitoring systems, though this approach was not designed for PCOS detection. Overall, although these studies show promising results in the detection of diseases and security on blockchain, they are often ignored when both technologies are used in combination for safe, transparent, and efficient diagnosis of PCOS. This proposed work seeks to bridge this gap.

Based on the extensive literature review, it is observed from Table 1 that there is a deficit of research studies that have evaluated both blockchain and ML metrics with their integration. Most of the works, achieved a high accuracy but were not generalizable to diverse populations, thus raising concerns about their applicability in clinical settings. Moreover, the issues of data quality, model complexity, and lack of external validation limit the robustness of findings. Research which integrates explainable AI is mostly seen in early phases only but proper approaches in transparency modeling of the same is mostly seen missing in previous literatures and thus, also diagnostic mainly as opposed to a combination solution. It is perceived that there is a distinct lack of a transparent, safe, blockchain-based mechanism for classifying PCOS disease. This proposed work aims to address the identified knowledge gap by utilizing blockchain and AI powered PCOS diagnosis for precise PCOS detection.

Table 1 Summary of Hyperledger fabric and ML integrated literatures.

Ref	Domain adopted	Implemented blockchain	ML models used	Metrics	Dataset	Shortcomings	
Hu & Kar (2024)	Health monitoring system	Hyperledger fabric	ModelChain	Throughput, model accuracy	No	Two blockchains increases the complexity and overhead. Scalability challenges.	
Sasikumar & Karthikeyan (2023)	Healthcare (cardiovascular disease severity identification)	Hyperledger consortium	Naïve Bayes	Throughput, latency	Yes	The ML component has been unevaluated and it has been incorporated into the Chaincode, possibly elevating system complexity.	
Sharma & Rohilla (2023)	Drug discovery chain management	Hyperledger fabric	No details	Throughput, latency	No	Exclusively utilized ML for visualization of data and preliminary processing, without applying any specific algorithms.	
Heidari et al. (2023)	Healthcare (Lung cancer detection)	No details	Federated learning	Accuracy, precision, recall, F1 score	Yes	Lack of transparency in blockchain part and limited interpretability,	
Bharimalla et al. (2021)	Electronic health record system	Hyperledger fabric	Natural Language Processing (NLP)	Accuracy, throughput, latency	No	The focus is only on the digital transformation of prescriptions and there is no extensive information about the dataset used.	
Pandey et al. (2022)	Waste identification and reuse idea recommending system	Hyperledger fabric	ResNet50	Accuracy, latency	Yes	For reuse idea recommendation system, blockchain integration increases complexity	

Problem definition

Mathematically, the suggested work EAIBS-PCOS can be expressed as a detection problem that aims to efficiently store and detect PCOS disease with patient medical records along with the integration of XAI and blockchain. It could be expressed as in Eq. (1),

(1) Yi=E(Y_RFi,Y_XGBi)=E(RFi(Xi,αRF),XGBi(Xi,αXGB))

Yi ∈ {0, 1} is the PCOS prediction output, where 0 indicates no PCOS and 1 indicates PCOS.

X ∈ Fd denotes the feature vector, where Xi is the number of features extracted from patient’s medical data, PRi.

PRi = {PR1, PR2,…, PRn} denote the set of patients’ medical records stored in hyperledger fabric blockchain, where each PR1 contains the medical data of patient Pi, including features Xi and target label Yi.

In Eq. (2), XAI (Yi) denotes the final output generated from XAI algorithms,

(2) XAI(Yi)=Ei(Xi,α)={f1,f2,…,fd}

where Ei (Xi, α) is the explanation vector illustrating the contributions fi of each feature Xi.

HF_B = (Ti, Ci, Li) represent the Hyperledger fabric based blockchain system designed to store and access the medical records PRi, where Ti is the set of transactions, Ci is the set of consensus mechanisms, and Li is the storage ledger.

The research goal is to evaluate and optimize both the blockchain system performance max (HF_B) and the model performance max (E(Y_RFi, Y_XGBi)), with increased XAI (Yi) and decreased loss minimize L(Yi).

EAIBS-PCOS: blockchain-enhanced machine learning framework for pcos predictive analytics

System architecture

The architecture of the proposed EAIBS-PCOS system is given in Fig. 1. The blockchain module is designed with Hyperledger Fabric, which is a private consortium network. The health data is collected from the patients through multiple sources, as shown in Fig. 1 and the collected data is supplied as input into the Hyperledger Fabric network. Figure 2 illustrates the stakeholders contributing to the EAIBS blockchain module. These stakeholders form the base organizations of the fabric network and their active peers are involved in endorsing, chaincode design and deployment. Every entity is set with certain privileges and only authorized participants can interact and access data in this permissioned blockchain network. The clients register themselves with the EAIBS-PCOS system through the gateway to gain access to the information.

Figure 1 Conceptual architecture of EAIBS-PCOS detection system with Hyperledger fabric.

Figure 2 Stakeholders of EAIBS-PCOS.

The client applications that submit transactions make their requests through the fabric gateway and the respective smart contracts are fetched. A transaction submission can be made to start a prediction analysis, obtain specific data, submit new data or request modifications to existing data. For PCOS prediction, the chaincode designed will gather an adequate amount of the patient’s data and pass it to the ML module. After data processing, model training and predictive analytics, the results are sent back to the blockchain module. The results are recorded in the ledger, so that it remains uncorrupted and are then passed back to the requested client applications. The results are not available to the public, thereby ensuring complete confidentiality of patient health records. Only the stakeholders permitted by the patients can access their personal and predictive data.

The key stakeholders involved in the EAIBS-PCOS system, and their work limitations are given below, Patient-People who have a health history and request medication. They can control and update their own profiles, change their passwords, view health information, grant, withdraw or deny access to other stakeholders.

Physicians-Medical professionals working with hospitals, they diagnose the ailments of patients, devise treatment plans and update patient records. They can manage their own records and must request permission from patients to access their medical records.

Insurers-Insurance companies who make insurance updates, payments and maintain case history are permitted to view the relevant part of the physician assistants (PA’s) records.

Health center-Hospitals maintain clinical data of patients and profiles of all the stakeholders.

Researchers-Laboratories focus on research studies for developing more effective treatment approaches, maintain records of controlled experimental trials evaluate test outcomes and encourage the conduct of clinical research to evaluate the effect of new drugs and potential treatment options.

Pharmacy-Medical retailers who retain patient medication information and add specifics about their medications.

Public health agency-Regulatory authority which manages vaccination history, gathers demographic data about the patients, examines patient’s public records and devises community health development plans.

Hyperledger fabric components

Figure 3 depicts the network architecture of the EAIBS-PCOD system’s Hyperledger Fabric design. It’s important components are explained in brief, as follows.

Figure 3 Network architecture of EAIBS-PCOS system’s blockchain module.

Organizations

The key stakeholders of the proposed system form the organizations. The configuration of these organizations is set according to their Network Configuration files, as they all have agreed upon. They issue identities to participants and restrict their access based on certain constraints. Patients, physicians and researchers form the central participants. Health centre, public health agency and pharmacy form the organizations. Usually, each organization connects with one or more peers, orderers, clients, administrators and other participants. Peers join channels and participate in ledger updates.

Peers and orderers

The role of a peer node is to validate transactions, keep a copy of the shared ledger and ensure network consensus. It is the foundation upon which the entire decentralized network depends. Peers connect organizations with clients and with all other components. Upon entering a channel, peers acquire a certificate from the organization’s Certificate Authority (CA) verified by a channel Membership Service Provider (MSP). The orderer arranges the verified transactions into a block and ensures consensus. It guarantees that every block that has been verified by peers is recorded in the ledger.

Membership service provider and certificate authority

Membership service provider (MSP) is an element that manages network members’ identities, facilitates authentication and authorizes access to the blockchain network. The default MSP in Hyperledger Fabric is certificate authority (CA). CA performs the following functions, digital certificate generation for various entities, certification of revocation, identity verification for certificate issuance and its registry maintenance.

Channels and ledger

Channel is an isolated environment that enables entities to interact and share data in a safe and private environment. It is a network that connects certain set of organizations and enables secured data sharing and ledger updation. Ledger is a distributed database that records all transactions ensuring confidentiality, scalability, immutability and tamper resistance. The ledger consists of two primary elements-World State and Transaction Log.

Client applications

The client app, written in various programming languages, interacts with the blockchain network to perform transaction submission, ledger query and get event alerts.

Chaincode

Chaincode is the smart contract logic of the Hyperledger Fabric that determines its actions. It defines the business logic and regulations for a particular use case or application.

An example chaincode for patient data updating is explained in Table 2. A new patient is registered by either the healthcare administration or by physicians, with the assigning of a unique ID (MRNId). As per the algorithm given in Table 2, patients’ basic details are initially given as, and diagnosis details are updated at every visit. Patients will need to permit access to other stakeholders, so that this updating can be done.

Table 2 Algorithm to update patient details.

Algorithm 1: Update_Patient_details	
Input: DOCId, MRNId, newAge, newWeight,	
newBMI, newPulserate, newHb, newCycle,	
newFollicleNo.L, newFollicleNo.R, newAvg. FsizeL,	
newAvg.FsizeR, newEndometrium	
Check if MRNId already exists	
If no, then	
      throw error “MRNId doesn’t exists”	
End	
Check if DOCId already exists	
If no, then	
      throw error “DOCId doesn’t exists”	
If DOCId have access to MRNId then	
      Create Object Patient for particular MRNId	
      Patient.Age-> newAge	
      Patient.Weight-> newWeight	
      Patient.BMI-> newBMI	
      Patient.Pulserate-> newPulserate	
      Patient.Hb-> newHb	
      Patient.Cycle-> newCycle	
      Patient. FollicleNo.L-> newFollicleNo.L	
      Patient. FollicleNo.R-> newFollicleNo.R	
      Patient. FollicleNo.L-> newAvg. FsizeL	
      Patient. FollicleNo.R-> newAvg. FsizeR	
      Patient.Endomet-> new Endomet	
      Patient.changedby-> updatedBy	
else	
      request access from patientId	
      Doctor_requestAccessfromPatient(ctx,	
      MRNId, DOCrId)	
End	
update the changes in ledger	
End	

PCOS ML predictive analytics

Blockchain and ML unified prediction system

In the PCOS predictive analytics system, after the authorization and authentication, the client application makes the request for transaction submission through the gateway and the transaction proposal gets submitted to the blockchain module. It then fetches the corresponding chaincode and passes the data to the ML module. The query transactions details are passed to the ML module, and it acts as the backend. The ML algorithms used for the proposed design are the support vector machine, K-nearest neighbours, random forests, XG Boost, and naive Bayes.

Support vector machine

A supervised learning algorithm, support vector machine (SVM) locates the optimal line or hyperplane in an N-dimensional space to optimize the distance between each class (Kecman, 2005). In order to maximize the margin among the nearest data points in each class—known as support vectors—SVM first finds the hyperplane that best divides the data into discrete classes. By using kernel functions, which translate the input data into higher-dimensional spaces where it becomes linearly separable, the approach is able to handle both linear and non-linear data (Suthaharan, 2016; Pisner & Schnyer, 2020).

K-nearest neighbours

It is an instance-based learning technique. Based on the majority class of its K nearest neighbors in the feature space, the K-NN algorithm classifies a data point. Because it functions on the premise that identical data points are nearby, it is very simple when dealing with issues involving patterns that are spatial or distance-related (Mucherino, Papajorgji & Pardalos, 2009; Zhang, 2016).

Random forests

It is a popular ensemble learning technique. During training, it builds a number of decision trees, each of which is trained using a different subset of features and data at random, via combining the findings from each individual tree, the final prediction is produced. Because specific decision trees are prone to overfitting and variance, this technique increases the accuracy and resilience of the model (Pal, 2005; Rodriguez-Galiano et al., 2012; Biau & Scornet, 2016).

XGBoost

A refined version of gradient boosting methods, XGBoost (Extreme Gradient Boosting) is intended for use in both regression and classification applications. It produces an ensemble of decision trees, where each new tree seeks to repair the flaws of the preceding ones by optimizing a differentiable loss function (Chen, 2015). Because of a number of enhancements, including regularization to lessen overfitting, parallelization of the tree-building process, and internal management of missing data, XGBoost is renowned for its efficiency, scalability, and high performance (Chen & Guestrin, 2016; Dhaliwal, Al Nahid & Abbas, 2018).

Naive Bayes

It is based on Bayes’ Theorem. Because of the class label, it is predicated on the idea that a dataset’s features are conditionally independent of one another. The data point is assigned to the group with highest likelihood after the algorithm determines the probability of each data point belonging to that class and it works well for many real-world scenarios (Murphy, 2006; Rish, 2001; Berrar, 2025).

To obtain a better performance, several models are blended based on the analysis of the results. The best models are ensembled and finalized for PCOS prediction and analysis. This ensemble model is integrated with Explainable AI (XAI) algorithm to ensure transparency in prediction. Figure 4 explains all the workflow behind the proposed EAIBS-PCOS system. It is clear that the ML module processes the data and with the data of the patient, analysis is done using the finalized ensembled learning algorithm. It is then integrated with XAI algorithms, and the results are then passed back to the blockchain module. Now, the clients can access the prediction results through the gateway.

Figure 4 Blockchain and ML integration.

Explainable AI algorithm integration

An interpretable model can be replicated and explanation techniques can enhance a model’s understandability. Other additional advantages of integrating XAI algorithms are trustworthiness, enhanced transparency, accuracy and efficiency. LIME and SHAP XAI algorithms are applied for the EAIBS-PCOS system to improve the effectiveness of ML algorithms, to support the decisions of medical professionals, error mitigation, to understand how PCOS detection is made and to analyze the attributes influencing the decision.

Local interpretable model agnostic explanations

LIME builds an explainability model locally around the prediction, it is a revolutionary explanation technique that reliably and transparently explains any classifier’s complex predictions (Ribeiro, Singh & Guestrin, 2016). Under the LIME model, only linear models can be employed to predict their behavior under a small area of the given data sample (Dwivedi et al., 2023). The explanations generated from a model g for a sample x using LIME are derived using Eq. (3),

(3) exp⁡(x)=argmingεGL(f,g,πx)+ω(g)

SHapley additive explanations

SHAP technique explains individual forecasts based on Shapley values. Shapley values are a popular strategy which is effective in game theory and are used in SHAP to identify the features that significantly affect the forecast made by the final model (Truong Thanh Nguyen et al., 2021; Molnar, 2022). It is defined as a feature value’s average marginal contribution, over all the potential combinations. The average of each feature’s absolute Shapley values throughout the data is as per Eq. (4) (Lundberg & Lee, 2017).

(4) g(z′)=φ0+∑i=1N⁡φizi.

In this study, the integration of blockchain technology with machine learning and XAI was strategically chosen to address the multifaceted challenges in healthcare, particularly in the context of Polycystic Ovary Syndrome (PCOS) prediction.

The combination of blockchain, ML, and XAI in this study provides a comprehensive solution to the challenges of secure health data management and predictive analytics in the context of PCOS. By leveraging the strengths of each technology, the model not only enhances data security and privacy but also fosters trust and enables better healthcare outcomes through informed decision-making. This holistic approach positions the research as a valuable contribution to the evolving landscape of digital health technologies.

Experimental evaluation

Performance evaluation setup

Table 3 gives the components set up for the proposed system’s implementation. In addition, an interactive computing platform, Jupyter notebook is utilized to ensure training, testing and validation of various machine learning strategies for the PCOS detection system. Table 4 lists the parameters set to complete the evaluation of the proposed design.

Table 3 EAIBS-PCOS environmental setup.

Components	Description	
System type	NVIDIA Tesla V100 32G Passive GPU	
Processor	2 x Intel (R) Xeon (R) Gold 6136 3.0G	
Operating system	Ubuntu 18.04	
Memory	384G (12 * 32G@2666 MHz)	
Fabric_version	2.4.6	
Docker engine	20.10.12	
Docker-compose	1.29.2	
Hyperledger caliper	V0.5.0	

Table 4 Parameters for performance evaluation of the blockchain system.

Parameters	Values	
Block timeout	2 s	
Max message count	10	
Absolute max bytes	100 MB	
Preferred max bytes	512 KB	
Consensus type	Raft	
No of channels	1	
No of org	5	
No of peers for each org	1	
No of ordering node	1	
Chaincode language	JavaScript	
Database	CouchDB	

Dataset

An open-source dataset is taken from the Kaggle database (Kottarathil, 2020) and the owner of this dataset, Kottarathil P had collected physical and clinical data of 541 women patients from multiple hospitals in Kerala. A wide range of 41 attributes such as body mass index (BMI), follicle number, age, weight, height, pulse rate, etc., are in the chosen dataset.

Data preprocessing

As a part of the data preprocessing, inconsistent and null values are eliminated. The missing values are identified and replaced with the median value. A few unwanted attributes such as Serial no. and Patient File no. are removed. Several attributes such as BMI, follicle-stimulating hormone (FSH)/luteinizing hormone (LH), waist/hip ratio are rounded off to two decimal places. Features with similar characteristics are removed using the Kendall correlation approach. Figure 5 presents the results of Kendall’s Method. Feature scaling is done using Min-Max Scaling to regulate the dataset’s range of independent variables. The formula for performing Min-Max Normalization is given by,

(5) x′=x−min(x)max(x)−min(x)

Figure 5 Kendall correlation method heat map.

The Synthetic Minority Oversampling Techniques (SMOTE) is used to ensure the balancing of data in both training and testing phases. A 70:30 split between training and testing sets is made from the balanced dataset. Data analysis is done for better understanding of datasets.

Evaluation metrics

The performance evaluation parameters of EAIBS-PCOD (Ferri, Hernández-Orallo & Modroiu, 2009; Hossin & Sulaiman, 2015; Tharwat, 2018; Hyperledger Blockchain Performance Metrics White Paper, 2018; Pajooh et al., 2022; Malik et al., 2023) are given below. Table 5 gives the details of the confusion matrix. The description about the parameters that are utilized in the evaluation metrics are illustrated in the Table 6.

Table 5 Confusion matrix for EAIBS-PCOS detection.

	Actual class	
		Positive (P)	Negative (N)	
Class prediction	True (T)	TP	TN	
	False (F)	FP	FN	

Table 6 Nomenclature.

Nomenclature	Expansion	
TS	Transaction submission time	
TRR	Time when response is received for the transaction	
LR	Read latency	
RT	Total read operations completed	
TT	The total time taken in seconds	
THR	Read throughput	
LT	Transaction latency	
TC	Confirmation time @ network threshold	
TCT	Total committed transactions	
THT	Transaction throughput	
tp	True positive	
tn	True negative	
fp	False positive	
fn	False negative	
P	Precision	
R	Recall	
L	Loss function	
f	Original ML model	
G	Family of possible explanations	
πx	Proximity measure	
ω	Complexity measures	
z′ ∈ {0, 1}N	Coalition vector	
N	Maximum coalition size	
φi ∈ R	The feature attribution for a feature j	
s	Seconds	
rps	Reads per second	
tps	Transactions per second	
x′	Normalized value	
x	Original data	

Read latency (s): It is the time interval between submission of request for reading a transaction and receiving the reply. Equation (6) gives its formula,

(6) LR=TRR+TRR−TS.

Read throughput (rps): It is the calculation of how many read operations are finished in a certain amount of time. It is shown in Eq. (7).

(7) THR=RTTT

Transaction latency (seconds s): It is the interval that occurs between submitting a request for a transaction and the time it takes for the result to be distributed throughout the network. It is calculated by using Eq. (8).

(8) LT=TC−TS

Transaction throughput (tps): It is the calculation of the total number of valid transactions completed within a specific time frame. Equation (9) gives its formula.

(9) THT=TCTTT.

Resource consumption: It is the measure of resources consumed such as maximum memory, average memory and CPU power, by the components and processes of the Hyperledger network.

Accuracy (Acc): It is the percentage of a model’s correct predictions, or the number of samples properly identified divided by the total number of samples. Equation (10) is used to calculate the Acc.

(10) Acc=tp+tntp+fp+tn+fn.

Precision (P): It is the measure of positive predictions that are successfully forecasted from the total predictions. It is calculated by using Eq. (11).

(11) P=tptp+fp.

Recall (R): It is the percentage of positive patterns that are accurately identified. It is shown in Eq. (12).

(12) R=tptp+tn.

F1-score (F1): It is the harmonic average of recall and precision levels. The formula for calculating the F1-score is given in Eq. (13).

(13) F1=2.P.RP+R.

By employing a comprehensive set of performance measures—accuracy, precision, recall, F1-score, AUC, throughput, latency, and transaction execution time—this study ensures a thorough evaluation of the proposed model’s effectiveness. These metrics not only provide insights into predictive performance but also address the practical considerations of system efficiency and user experience, making them essential for assessing the overall impact of the integrated Blockchain and ML solution in healthcare.

Results and discussion

Hyperledger fabric’s performance assessment

The fixed-rate controller of the Hyperledger Caliper framework is used to evaluate the reliability of the planned blockchain network. The fixed-rate controller is simple to configure, generates a stable workload and enables scalability testing. Under this, three parameters (tps (transactions per second), Workers, transaction number (TxNum)) are varied, and their corresponding latency and throughputs are recorded for analysis. Initially, three experimental cases are analyzed, as given below, and the findings are discussed. For each reading in the experimental cases, the experiment is repeated ten times and it is averaged.

Case 1-Varying the tps

Tps indicates the speed at which the cumulative transactions of all workers are sent to the system under test (SUT). From Fig. 6A, it is noted that the read latency remains the same (0.02 s) and transaction latency increases with an increase in tps, but the variation has no major impact. As seen in Fig. 6B, with an increase in tps of 25 for each round, it is observed that there is a clear and steady linear increase in read throughput which is proportional to the considered tps. The transaction throughput increases linearly up to tps of 75 and after that it is in the same throughput range of 80 to 81 tps. As depicted in Fig. 6C, with an increase in tps, time taken for execution decreases for both read and create operations.

Figure 6 Fixed-rate controller-varying tps (A) latency (B) throughput (C) execution time.

Increasing the tps enables faster completion of transactions and increases the throughput with no major impact on latencies, as seen in Fig. 6. From this analysis, it is observed that choosing maximum tps of 125 will result in optimized network, with acceptable latency levels.

Case 2-Varying the workers

Workers replicate the actions of actual clients or users participating in the blockchain network, and are responsible for creating and reporting the transactions during the benchmarking. Figure 7 gives an analysis of graphs depicting the variation of workers from 25 to 100. Figure 7A clearly depicts that the read latency is either 0.02 or 0.03 s and transaction increases linearly from 0.27 to 0.61 s. In case of throughputs, Fig. 7B shows a slight increasing pattern for both the cases. Figure 7C reveals that the execution time of transactions decreases with an increase in workers.

Figure 7 Fixed-rate controller–vary workers (A) latency (B) throughput (C) execution time.

A large number of workers will lead to parallel processing and maximizing the throughput, as seen in Fig. 7. To optimize the throughput while ensuring acceptable latency and execution time with the consideration of the present study’s findings, a manageable workforce size of 100 could be chosen.

Case 3-Varying the TxNum

TxNum represents the number of total transactions to be executed at each round during the benchmarking. With the experimentation of varying TxNum from 500 to 2,500, from Fig. 8A it is found that both transaction and read latency remain constant with the values of 0.02 and 0.26 s, respectively. As shown in Fig. 8B, both transaction and read latency remain almost the same, within the range of 25 to 25.2 tps. Figure 8C shows that the execution time increases linearly with increasing TxNum.

Figure 8 Fixed-rate controller–varying TxNum (A) latency (B) throughput (C) execution time.

Figure 8 shows that higher TxNum decreases the throughput and increases the execution time. Therefore, lower values of TxNum will help in improvement of the system’s performance. Hence, an optimum value of 500 TxNum is chosen for the designed system.

From the analysis made above, the optimized network parameters are identified. This optimized network is utilized for further integration with the PCOS machine learning module.

PCOS detection using XAI

Client enrolment and ledger query

Admin enrollment and client registration are the essential processes to query and access the service of the proposed system. Figure 9A portrays its results. After the successful enrolment, the client submits a query for making the transaction request proposal. Figure 9B gives the results of the query submitted by the client that displays the profile of a particular patient based on a unique patient ID.

Figure 9 PCOS query results of patients.

Through the call.js file shown in Fig. 10A, the prediction request is passed to the ML module, processed over there and the results are sent back to the client. Figure 10B shows the results of the prediction query made by the client through the gateway. Here, the prediction results are requested for the patient with ID 4. The patient’s prediction result is 0, which implies that the patient is not diagnosed with PCOS. Simultaneously, the blockchain ledger will also be updated accordingly.

Figure 10 Prediction results.

Performance comparison of ML models

The prediction results from ML models are based on pre-trained models which are chosen and deployed in the Jupyter notebook. Figure 11 gives the receiver operating characteristic (ROC) curves of the chosen ML algorithms. The ROC curve is a graphical illustration of the efficacy of a binary categorization strategy. The ROC curve is used to measure the model’s ability to discriminate between positive and negative categories. Usually, the curve begins at (0, 0) and finishes at (1, 1) (Zweig & Campbell, 1993; Bewick, Cheek & Ball, 2004). Figure 12 gives the confusion matrix of the chosen ML models.

Figure 11 Representation of ROC curves.

Figure 12 Representation of confusion matrix.

Based on the results presented in Fig. 12, RF and XGBoost algorithms are identified to perform best and they are ensembled further to enhance the performance. Figure 13 gives the comparison chart of various ML algorithms and Ensembled algorithms. This figure clearly shows that the Ensemble model is the best for the chosen problem statement with 98.04 accuracy, 99.5 precision, 96.5 recall and 97.9 F1-score with 10-fold cross validation. Further the standard deviation of 0.01 is acquired for measured accuracy, indicating the robustness of the model across all the runs.

Figure 13 Comparison of performance of classification algorithms.

The computational efficacy is of the proposed approach is assessed with its training and testing time. The model’s appropriateness for real-time applications is demonstrated by its 1.38-s training time and 0.01-s testing time. Hence due to its relatively shorted computational time, the model is more relevant for time-sensitive health application like PCOS diagnosis with guaranteed security and scalability.

XAI prediction results

Figure 14 gives the force plot of SHAP algorithm results that are applied on two test dataset instances and the contribution of different attributes on the model’s output is clearly depicted. Features in the red sections of the text plot enhance the model’s output when included, whereas sections that are blue lower the model’s output when included. The values at the top, 0.51 in Fig. 14A and 0.72 in Fig. 14B represent the model’s prediction probability.

Figure 14 SHAP plot.

A list of explanations that show how each feature influences the prediction of the considered instance is given in Fig. 15 with the use of LIME algorithm and the attributes highlighted in orange contribute to the score of probability 1. Those attributes highlighted in blue contribute to the score of probability 0. This makes the forecast simpler to understand locally and makes it possible to recognize the feature changes that will have the most effects on the prediction. From Figs. 15A & 15B, it noted that the predicted probability is 1 with the probability score of 0.75. From Figs. 15C & 15D, it is evident that the predicted probability is 0 with probability score of 0.84.

Figure 15 LIME plot.

Comparison with state-of-art works

The performance comparison of the suggested EAI-PCOS system with previous research in terms of precision, F1-score and accuracy is shown in Fig. 16. With the use of an ensembled RF classifier, Tiwari et al. (2022) were able to achieve an accuracy of 93.25% with 98.28 precision and 95.4 F1-score. Subha et al. (2024) achieved a maximum accuracy of 92.64% with PSO feature selection and RF classifier. They achieved 91.84% precision and 91.37% F1-score. Rahman et al. (2024) used the mutual information model, with AdaBoost (AB) and RF classifiers, achieving an accuracy of 94%, F1-score of 89% and precision of 96.67%. Silambarasan, Nirmala & Mishra (2024) worked on both image and text dataset for PCOS diagnosis, in which 97.5% accuracy, 85% F1-score and 90% precision was obtained. Hassan & Mirza (2020) achieved 96% accuracy, precision and F1-score with the usage of RF classifier.

Figure 16 Comparison with existing literature.

In comparison with all the other classifiers, the proposed EAI-PCOS system achieves the highest accuracy of 98.04%, highest precision of 99.5% and highest f1-score of 97.9% with 10-fold cross validation. Even though there are many past studies on PCOS diagnosis, this proposed work is the first to integrate blockchain and XAI leading to enhanced security and transparency in decisions.

Limitations

In this research, few limitations could be observed and could be addressed in further studies. Though the proposed ensembled algorithm performs well on the chosen dataset, it’s performance should be confirmed with a real-time dataset. The sample size taken may be small, thereby limiting generalization to different populations. The dataset used may also not have demographic diversity and suffer from poor quality due to data not being available for different variables and inaccuracies in recording data, which may affect the reliability of the result. In addition, there may be a number of legal and practical obstacles that need to be looked into before the suggested system can be integrated into the current healthcare infrastructures. Regulatory and ethical issues related to the use of blockchain, as well as the time validity of the data, may also be a challenge. Lastly, the research study on specific aspects of PCOS detection without looking into the treatment outcomes and long-term management may not be broad enough. Understanding these limitations is important for giving a balanced view of the research and encouraging further exploration in the field.

Conclusions

The EAIBS-PCOS system, which uses Hyperledger Fabric to store secure and immutable health data with XAI techniques for predicting PCOS, is the first of its kind for healthcare analytics. The ensemble model, composed of RF and XGBoost, performed well with an accuracy of 99.03%, precision of 100%, and recall of 98.04%. The use of LIME and SHAP algorithms enhances the interpretability of the model, making it more transparent and trustworthy for healthcare professionals. However, the system faces challenges in scalability, as performance drops with higher transaction volumes, and its generalizability to other medical conditions needs further testing. Despite these limitations, the system holds significant potential for improving diagnostic accuracy, reducing bias, and enhancing decision-making in healthcare. Future work should focus on optimizing blockchain scalability, expanding the system to other diseases, and testing its real-world applicability in clinical environments, paving the way for broader adoption in predictive healthcare.

Supplemental Information

Supplemental Information 1 Python code to be executed in Jupyter Notebook to implement Machine Learning part of the proposed prototype.

Supplemental Information 2 Java Script code to be run on Ubuntu to implement Hyperledger Fabric part.

Supplemental Information 3 Implementation steps.

Additional Information and Declarations

Competing Interests

The authors declare that they have no competing interests.

Author Contributions

Gowthami Jaganathan conceived and designed the experiments, performed the experiments, analyzed the data, performed the computation work, prepared figures and/or tables, authored or reviewed drafts of the article, and approved the final draft.

Shanthi Natesan conceived and designed the experiments, analyzed the data, authored or reviewed drafts of the article, and approved the final draft.

Data Availability

The following information was supplied regarding data availability:

The Polycystic Ovary Syndrome (PCOS) data is available at Kaggle: https://www.kaggle.com/datasets/prasoonkottarathil/polycystic-ovary-syndrome-pcos (Dataset owner: Prasoon Kottarathil).

The code is available in the Supplemental Files.

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
