# Peer review of "Blockchain and explainable-AI integrated system for Polycystic Ovary Syndrome (PCOS) detection"

_PeerJ Computer Science, doi:10.7717/peerj-cs.2702_

## Round 0.1 · original submission · Major Revisions

All concerns of the reviewers should be taken in consideration while preparing a response and revising the paper.

Reviewer 1 ·

Basic reporting

All comments have been added in detail to the last section.

Experimental design

All comments have been added in detail to the last section.

Validity of the findings

All comments have been added in detail to the last section.

Additional comments

Review Report for PeerJ Computer Science
(Blockchain and Explainable-AI integrated system for Polycystic Ovary Syndrome (PCOS) detection)

1. Within the scope of the study, Polycystic Ovary Syndrome detection processes were carried out with both blockchain and explainable artificial intelligence for the secure storage of medical data.

2. In the introduction, what Polycystic Ovary Syndrome is, its importance, the development of its detection and its relationship with machine learning are mentioned in detail. In addition, the main contributions of the study to the literature are clearly stated.

3. In the Related works section, the use and integration of machine learning and blockchain in the literature, the place of deep learning and machine learning in Polycystic Ovary Syndrome detection studies and their importance in the literature are mentioned.

4. When the framework, system architecture, hyperledger fabric with various components and machine learning section proposed for Polycystic Ovary Syndrome detection are examined in detail, it is observed that it has the potential to make a significant contribution to the literature and contains a certain level of originality.

5. An open source dataset obtained from the Kaggle platform was preferred as the dataset in the study. When the literature is examined, although there are many different datasets that can be used for solving this problem, please explain why this dataset in particular was used and what its advantages/disadvantages are compared to other datasets.

6. Although the data preprocessing steps related to the dataset seem sufficient for this study, please state how the training/testing distribution was determined and/or whether different experiments were performed.

7. Various machine learning algorithms such as Support Vector Machine, Random Forests and Naive Bayes were used for the framework. Although there are different machine learning algorithms that can be used in solving this problem in the literature, the reason for choosing these models should be stated more clearly.

8. Using SHapley Additive Explanations and Local Interpretable Model Agnostic Explanations for Explainable Artificial Intelligence is both sufficient and has increased the quality of the study.

9. When both the types of evaluation metrics required for the analysis of the results and the results obtained are examined, it is observed that the study clearly affects its contribution to the literature.

As a result, the study has significant potential in terms of the subject addressed. However, attention should be paid to all the sections listed above.

·

Basic reporting

- There are already many research articles available on the similar topic what is the need of this paper? What is the motivation if this paper? What is the novel contribution of this article compared to other existing articles?
- The abstract should include the objectives, innovation and quantified results (generally).
-- The formal definition of the problem should be written in the introduction.
- Which drawbacks of the related state-of-the-art works have been addressed by the proposed method.
- Its hard to justify on the method because the proposed method should be explained in detailed and in step by step form regarding the flowchart.
- In the experiment system section, the platform, implemented codes, evaluation criteria, research questions and datasets should be explained in different subsections.
- The obtained results are not enough to justify the proposed method.
- Another real world workloads are required to evaluate the effectiveness of the proposed method.
- The utilized algorithms should be executed different time (at least ten times) and the obtained results of the data replication should be evaluated with the value of the standard deviation.
- The paper lacks a detailed comparison of the proposed method with other existing methods. The authors are suggested to compare the results with the recently published methods in terms of different related criteria.
- The conclusion and abstract are not in a same structure. The occlusion should support the finding and the demerits of the method.
- There is not enough explanation and discussion for the obtained results and figures of your experiments.
-statistical analyses, controls, sampling mechanism, and statistical reporting should be described in the manuscript.
- The limitations of the study were not clearly stated.
- A critical summary of the related works reviewed needs to be added at the end of this section with their methods used, findings, pros, cons, etc.
-The research gaps for this study need to be established first in the related work section
- What is the computation time for the algorithm? Provide running time for the proposed method? Provide the comparison of computation time between the proposed method and other works.
- There are many algorithm parameters in the proposed method. What's the influence of these parameters?

Experimental design

- The paper lacks a detailed comparison of the proposed method with other existing methods. The authors are suggested to compare the results with the recently published methods in terms of different related criteria.

Validity of the findings

There are many algorithm parameters in the proposed method. What's the influence of these parameters?

---

## Round 0.2 · accepted · Accept

The authors have satisfactorily addressed the reviewers' queries.

Reviewer 1 ·

Basic reporting

All comments have been added in detail to the last section.

Experimental design

All comments have been added in detail to the last section.

Validity of the findings

All comments have been added in detail to the last section.

Additional comments

Review Report for PeerJ Computer Science
(Blockchain and Explainable-AI integrated system for Polycystic Ovary Syndrome (PCOS) detection)

The responses to the reviewer comments and the changes made to the paper are sufficient. I recommend that the paper be accepted. Best regards.

·

Basic reporting

- I have reviewed the new version of the manuscript and consider that the authors have satisfactorily addressed the reviewers' comments. The manuscript has improved its quality. Also, its contribution to the state of the art is clear

Experimental design

- I have reviewed the new version of the manuscript and consider that the authors have satisfactorily addressed the reviewers' comments. The manuscript has improved its quality. Also, its contribution to the state of the art is clear

Validity of the findings

- I have reviewed the new version of the manuscript and consider that the authors have satisfactorily addressed the reviewers' comments. The manuscript has improved its quality. Also, its contribution to the state of the art is clear